# Real-Time Video Synopsis via Dynamic and Adaptive Online Tube Resizing

**DOI:** 10.3390/s22239046

**Published:** 2022-11-22

**Authors:** Xiaoxin Liao, Song Liu, Zemin Cai

**Affiliations:** 1Department of Electronic Engineering, Shantou University, Shantou 515063, China; 2Key Lab of Digital Signal and Image Processing of Guangdong Province, Shantou 515063, China

**Keywords:** video synopsis, collision cost, tube resizing, motion anti-facts

## Abstract

Nowadays, with the increased numbers of video cameras, the amount of recorded video is growing. Efficient video browsing and retrieval are critical issues when considering the amount of raw video data to be condensed. Activity-based video synopsis is a popular approach to solving the video condensation problem. However, conventional synopsis methods always consists of complicated and pairwise energy terms that involve a time-consuming optimization problem. In this paper, we propose a simple online video synopsis framework in which the number of collisions of objects is classified first. Different optimization strategies are applied according to different collision situations to maintain a balance among the computational cost, condensation ratio, and collision cost. Secondly, tube-resizing coefficients that are dynamic in different frames are adaptively assigned to a newly generated tube. Therefore, a suitable mapping result can be obtained in order to represent the proper size of the activity in each frame of the synopsis video. The maximum number of activities can be displayed in one frame with minimal collisions. Finally, in order to remove motion anti-facts and improve the visual quality of the condensed video, a smooth term is introduced to constrain the resizing coefficients. Experimental results on extensive videos validate the efficiency of the proposed method.

## 1. Introduction

Nowadays, a huge number of video cameras are used for security purposes for 24 h per day. These cameras produce an enormous amount of video material. The power of video over still images is the ability to represent all sorts of dynamic activities. Although the captured videos contain useful information for applications in security monitoring and criminal evidence, the storage, management, and control of huge amounts of recorded video are becoming more and more difficult. Efficient video browsing and retrieval are time-consuming due to the inherent spatio-temporal redundancies [1,2], where some time periods may have no activity or have activities that occur only in a small image region. Therefore, video condensation techniques have been widely investigated in the literature over the last decade. According to the different basic units of extracted condensed video, current video condensation methods can be divided into two categories: activity-based video synopsis and frame-based condensation techniques.

Frame-based video condensation aims to generate a short video that contains useful information by selecting frames with important content (e.g., moving objects). The generated video is usually composed of a set of representative video frames (also known as key frames) or video fragments (also known as key fragments) that have been stitched together in chronological order to form a shorter video. While frame-based video condensation is straightforward to implement, its performance is quite limited. These techniques lose either the temporal or semantic context of activities in surveillance videos [3].

Another popular approach to solving the video condensation problem is video synopsis. Unlike frame-based video condensation approaches, the activities of interest are shifted in the time domain to obtain a video representation that is more compact [4]. While the temporal compaction is made possible by compromising the chronological time between object tubes, the dynamics of each object are preserved in this process. Meanwhile, the synopsis video is also an index for the original one by pointing to the original time of each activity. Video synopsis techniques achieve higher efficiency than frame-based video condensation techniques because of their more detailed video analysis.

Placing different object tubes with the least collision in a limited video length is crucial to the performance of a video synopsis, as this can lead to the loss of important content and cause a chaotic viewing experience, which would decrease the efficiency and visual quality for surveillance applications. Displaying the maximum number of activities in one frame with minimal collisions means a greater computational complexity. Currently, the most common methods of online video synopsis are carried out by optimizing complicated energy functions [5], which have not evolved to meet the needs of surveillance applications.

In this paper, we propose a simple online video synopsis framework that considers collision reduction, tube resizing, and smooth stitching. The condensation ratio, collision cost, and visual quality of the condensed video are important in video synopsis technologies. In the proposed method, the number of collisions of objects is classified first; then, different optimization strategies can be applied according to different collision situations in order to maintain a balance among the computational cost, condensation ratio, and collision cost. If this is within an acceptable range, i.e., L<=RC[i]<H, the corresponding object will be scaled down to avoid collisions at time position *i*. In order to display more activities in one frame with minimal collisions, the tube-resizing coefficients are changed at different time positions. Therefore, a suitable mapping result, which is determined adaptively in optimization procedure, can be obtained in order to represent the proper size of an activity in each frame of the condensed video. Moreover, in order to remove motion anti-facts and improve the visual quality of the synopsis video, a smooth term is adopted to constrain the resizing coefficients.

The rest of this paper is organized as follows. Section 2 describes the related work on video condensation methods, emphasizing their novelty and contributions to the field. Section 3 explains the proposed online video synopsis method. The experimental details of the proposed method and the corresponding results are presented in Section 4. We finally discuss the conclusion in Section 5.

## 2. Related Works

Video synopsis is a popular approach to solving the video condensation problem, and it provides activity-based video condensation instead of frame-based techniques, such as video fast-forward [6], video abstraction [7], video montage [8], and video summarization [9]. The video fast-forward method introduced in [6] can skip some unnecessary frames. However, it is easy to lose frames with important information, such as fast-moving objects. N. Petrovic et al. [10] adopted an adaptive fast-forward approach in order to decrease the loss of fast activities. After that, some criteria for extracting key frames were proposed [11]. Generally speaking, frame-based video condensation is straightforward to implement through the extraction of key frames, although the short videos that are generated lose activities’ temporal and semantic context, which is important in video browsing and retrieval.

The purpose of video synopsis is to provide a short video representation while preserving the essential activities in the original video. Activity-based video condensation was first proposed by Rav-Acha et al. [12,13,14] under the concept of video synopsis, in which most of the activities in the original video are condensed into a shorter period by simultaneously showing multiple actions, even if they originally occurred at different times. The process of video synopsis includes two major phases: (i) online activity generation and storage in a tube; (ii) a response phase for generating a short video on the basis of background generation [15], tube rearrangement, and object stitching. The first phase belongs to the tasks of video object segmentation [16,17], moving object detection [18], and tracking in computer vision. The background extraction algorithm ViBe that was adopted in the online video synopsis framework was introduced in [15]. It stores a set of values for each pixel at the same location or in the neighborhood. Then, ViBe compares the set to the current pixel value in order to determine whether that pixel belongs to the background and adapts the model by randomly choosing which values from the background model to substitute. Finally, when the pixel is found to be part of the background, its value is propagated into the background model of a neighboring pixel. While the background of the original video is complex, deep-learning-based object-tracking methods can be used to generate object tubes [19].

Optimization of the energy function is the most important part of video synopsis, and it aims to find the best rearrangement of extracted tubes in order to display most of the activities in the shortest time period. In order to reduce collisions and increase the condensation ratio, tube rearrangement is decided by the optimization module, and it guarantees the minimal cost for each of the tubes considered. Many spatio-temporal analysis methods have been proposed. Optimization approaches can be further divided into two classes: online and offline. Most video synopsis methods employ offline optimization for all generated tubes in order to find the global optimum. Xu et al. formulated the optimization problem of activities in terms of set theory and adopted the mean shift to solve it. However, temporal consistency was not considered in the tube rearrangement in this study. It was found that the particle swarm algorithm (PSO) could reach the global minimum solution, but with a lower computational cost. The PSO was used to solve the energy minimization function and to generate a synopsis video [20]. However, offline optimization methods are complicated and time-consuming. In recent studies, online optimization that applied rearrangement with each new activity was increasingly employed to find the local optimum. Online video synopsis enables tube rearrangement at the time of detection without any wait before starting optimization [21,22]. Huang et al. [23] emphasized the importance of online optimization techniques and proposed a maximum a posteriori probability (MAP) estimation in order to decide on the temporal locations of an incoming tube without prescreening the entire video. However, their proposed method completely ignored activity collision situations in order to improve the runtime performance. Activity collision is still the biggest challenge in video synopsis. Ruan et al. [24] proposed a novel graph-based tube rearrangement approach for online video synopsis in order to reduce collisions between tubes. Solutions such as simulated annealing [25], fuzzy C-means aggregation [26], and minimized sparse reconstruction [27] were also used to rearrange different tubes during optimization. Li et al. [28] proposed a video synopsis technique in order to decrease the collisions between moving objects’ tubes. The tubes were rearranged in the temporal domains, and the sizes of the objects were scaled down if a collision was detected. A metric representing the down-scaling factor of each object was used in the optimization step. However, for each tube, the metric was constant; then, the down-scaling operation decreased the visual quality and produced more block anti-facts that might disturb the user in surveillance applications.

## 3. The Proposed Method

In this section, we will introduce the proposed online video synopsis framework in detail, which considers various issues such as collision reduction, tube resizing, and smoothing. Using this framework, a suitable mapping result can be obtained in order to represent the proper position of a newly generated tube in the synopsis video. Meanwhile, the resizing coefficients of the corresponding tube, which are dynamic in different frames, can be adaptively determined.

### 3.1. The Optimization Framework

In traditional offline video synopsis, objective functions always consist of complicated and pairwise energy terms that involve a time-consuming optimization procedure. Therefore, a simplified objective function according to the characteristics of online video synopsis is first defined. Let *E* be the energy function and let *M* represent the mapping from the original video to the condensed video. Then, the energy function in the framework can be defined as
(1)$$E\left(M\right)=\lambda \cdot E_{c}\left(M\right)\cdot E_{r}\left(M\right)+E_{s}\left(M\right)$$
where $E_{c}\left(M\right)$ and $E_{r}\left(M\right)$ represent the collision cost and the cost due to the tube resizing, respectively. λ is a weighting factor. Obviously, the purpose of tube resizing is to offer possibilities for collision reduction. The balance between collision reduction and visual quality is important, which means that the resizing operation should be dynamic. $E_{s}\left(M\right)$ is another term for smoothing the scale variations of tubes between two frames in the synopsis video in order to suppress motion artifacts.

(1) **Collision Term $E_{c}\left(M\right)$:** In the condensed video, when a collision occurs between two tubes, the collision term is used to calculate the sizes of the collision areas of the entire regions in which collisions occur. For example, when tube *o* and tube *p* collide in the synopsis video, according to their positions in the original video, we calculate the size of the area of the intersection between the circumscribed rectangles of the corresponding foreground regions. The collision term can be formulated as
(2)EcM=∑o,p∈O∑o′∈O*,p′∈P*Recto′∩Rectp′
where Re*ct*(·) denotes the calculation of the size of the bounding rectangle’s area. *O* is the set of all current tubes and $O^{*}$ is a set of new tubes with a new time index and resizing factors that correspond to the mapping result of tube *o*.

(2) **Resizing Term $E_{r}\left(M\right)$:** In order to reduce the effects of collisions in the synopsis video, scaling down every object in a scene before rearrangement is an effective treatment. However, a down-scaling operation for a tube with a constant metric may decrease the visual quality and introduce more anti-facts in the condensed video. Scene changes are obvious for a tube that is temporally shifted in a video synopsis. In this study, a resizing term is introduced in order to penalize the cost of scaling down an object, and this can prevent excessive shrinking through minimization. The resizing term can be defined as
(3)ErM=∑xo∈X∑iMexpδxoi2·expηRectoi
where $x_{o}$ denotes the resizing coefficient of object *o*, and it is modified in a specific interval given by *X*. In our experiments, the interval is between 0.4 and 1, i.e., X=[0.4,1]. *M* denotes the tube’s length. The component $\exp\left(\frac{\delta }{{\left(x_{o}^{i}\right)}^{2}}\right)$ is defined in exponential form to prevent the resizing coefficient from closing to the left of the interval. When this happens, the component sharply increases to achieve a suppressive effect. Instead, when a collision between two objects occurs, it would be better if the size of the larger object could be reduced. The second component expηRectoi takes this situation into account. Suppose that the other variables remain the same; then, the larger area of $o_{i}$ achieves a smaller value of expηRectoi. Hence, this increases the possibility and ratio of shrinking for larger objects. In other words, a reduction in the size of smaller objects is more penalized than a reduction in the size of larger ones. δ and η are two tuning parameters.

(3) **Smooth Term $E_{s}\left(M\right)$:** In order to avoid motion anti-facts in the generated short video, another smooth term $E_{s}\left(M\right)$ is introduced in order to allow the resizing coefficients to vary systematically and smoothly in the temporal domain. This can be achieved by minimizing the second-order differences in the corresponding variables, which are expressed as
(4)$$E_{s}\left(M\right)=\sum_{x_{o}\in X}\sum_{j=2}^{M-1}{\left(2x_{o}^{j}-x_{o}^{j-1}-x_{o}^{j+1}\right)}^{2}$$

By incorporating object segmentation and tube extraction, a video synopsis framework that integrates the simplified energy function illustrated in Equation (Equation 1) is obtained. The online framework is shown in Figure 1.

There are three variables to be determined when minimizing the energy function E(M), i.e., the position variable, resizing coefficient variable, and speed variables. The energy function given in Equation (Equation 1) is a mathematical relationship in which the values of these dependent variables are determined. In order to achieve a condensed video with a high quality in real time, the controlled variable processing technique is employed. For example, we can keep the position variable and resizing variable constant (controls) during optimization according to the speed variable. At that time, the position variable and resizing variable are also known as constant variables or simply as ’controls’. After that, we minimize the energy function according to the resizing coefficient variable and control the other two variables, and so on. Based on the processing technique, the optimization problem is degraded into univariate function optimization problems. The graph-cut optimization method [29] or the simple simulated annealing approach [30] can be applied in order to solve the optimization problem. The procedure of optimization with simulated annealing is shown in Algorithm 1.
**Algorithm 1:** Procedure of optimization.
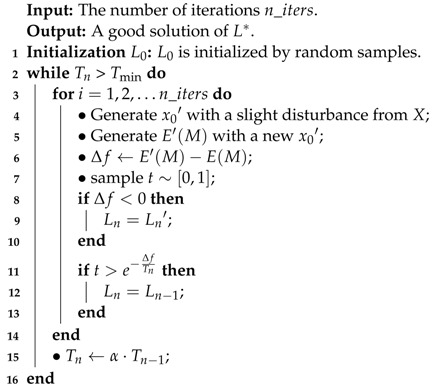


### 3.2. Implementation Details

A time label $t_{0}$ was assigned to a tube *T* in the tube buffer (also known as a tubelet) $G_{set}$ with a capacity of *P*. $t_{0}$ represents the time position at which the start frame of *T* appears in the synopsis video. When the time labels of each tube in the tubelet $G_{set}$ are different, the collision situation is also different. In addition, the collision of a tube is also related to the size of the object. Therefore, we adopted two thresholds, *L* and *H*, according to the number of collisions of a tube to choose a reasonable time label for the tube and a resizing factor for the object. A number of collisions that is less than *L* indicates that the collisions are within the acceptable range and that no adjustments are required. When the number of collisions is greater than *L* but less than *H*, we resize the corresponding object to avoid some of the collisions. The resizing coefficient ${\left(x_{o}^{i}\right)}^{*}$ in Equation (Equation 1) is adaptively determined through optimization. Otherwise, when the number of collisions is greater than *H*, we directly update the time label to be $t_{new}$, which means that the tube will be placed on the next frames of the existing condensed video. The main concern is that if the time label of tube *T* remains the same as that of $t_{0}$ in this situation, then the resizing coefficient of *T* will become very small and cause a significantly decreased visual quality in the synopsis video. In a word, there are three phases in the proposed online video synopsis algorithm, namely, the adjustment, addition, and subtraction phases.

**Adjustment phase:** As discussed previously, when the number of collisions according to the tube *T* in the container $G_{set}$ is between *L* and *H*, the dynamic resizing coefficients of the object in the temporal domain are determined and adopted to avoid collisions and have a guaranteed visual quality. The overview of the adjustment phase in the video synopsis is shown in Figure 2.

**Tube addition phase:** When a new activity tube $T_{new}$ is extracted from the original video, it is added to the container $G_{set}$. Then, the time label of $T_{new}$ is determined by traversing the length of $G_{set}$, which is updated by $T_{e}-T_{s}$, as illustrated in Figure 3.

For the tubes in $G_{set}$, all collisions are recorded in a vector RC, where RC[i] denotes the number of collisions that occur for $T_{new}$ at the time position *i*. After that, the smallest index *i* in RC is such that RC[i]<L is achieved and $T_{s}=i$. If no *i* such that RC[i]<L can be found, then the smallest index *i* such that L<=RC[i]<H is selected as a compromise. Otherwise, the tube $T_{new}$ is directly arranged on the next frame of the existing synopsis video. The processing of the new tubes is described in Algorithm 2.
**Algorithm 2:** Pseudocode for the addition of new tubes.
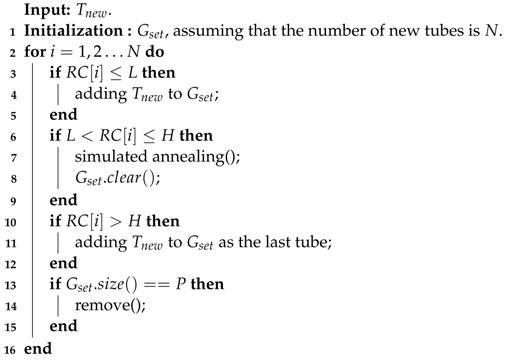


**Tube subtraction phase.** If the tubelet $G_{set}$ reaches the upper limit of its capacity *P*, the tube with the smallest time tag in $G_{set}$ is removed and rearranged in the condensed video. The tube subtraction processing reduces the computational time taken for the video synopsis in order to achieve real-time performance.

### 3.3. Temporal Consistency Constraints

Temporal consistency is another important problem in video synopsis; it aims to group related activities and display them together in the synopsis video in order to provide a better understanding of the scene for the user. In this work, a temporal consistency constraint was defined to maintain the time relationship between two moving objects. Let d(o,p,t) represent the Euclidean distance between two activities *o* and *p* in the *t*-th frame of the original video. The temporal consistency between *o* and *p* can be defined as
(5)$$D\left(o,p\right)=\exp\left(\frac{-\min\left\{d\left(o,p,t\right)\right\}}{\omega }\right)$$
where ω is a parameter for adjusting the distance metric according to the average size of the objects. In the computation, the adjustment parameter is set to ω=40. In addition, the relative temporal relationship between *o* and *p* is defined as
(6)$$R\left(o,p\right)=\exp\left(∥\left(t_{o}^{s}-t_{p}^{s}\right)-\left(t_{o^{*}}^{s}-t_{p^{*}}^{s}\right)∥\right)$$
where $t_{o}^{s}$ and $t_{p}^{s}$ denote the start times of two different activities, *o* and *p*, in the original video, respectively. Similarly, $t_{o^{*}}^{s}$ and $t_{p^{*}}^{s}$ are the start times of the corresponding objects $o^{*}$ and $p^{*}$ in the synopsis video. If activities *o* and *p* do not start at the same frame, another relative temporal relationship $R^{\prime }\left(o,p\right)=\left(t_{o}^{s}-t_{p}^{s}\right)\cdot \left(t_{o^{*}}^{s}-t_{p^{*}}^{s}\right)$ can be used to measure the relative temporal relationship between *o* and *p*.

## 4. Experiments

To evaluate the performance of the proposed video synopsis framework, many experiments were conducted on multiple surveillance videos. Firstly, a brief description of the surveillance video data used in our experiments is presented.

### 4.1. Experimental Setup

**Dataset:** Five surveillance videos were used in our experiments. The first (4514 frames) and the second videos (3950 frames) were provided by Nie et al. in [31] and Li et al. in [28], respectively. The other two videos were captured by surveillance cameras on the Shantou University campus. The third video with 9390 frames showed a road area with many pedestrians and vehicles. Moving objects in all directions appeared, and they moved across a wide range of areas. The fourth video, which had 970 frames, showed a corner of the campus, including pedestrians and cars whose trajectories were almost straight. More details about the dataset are given in Table 1.

**Evaluation Metrics:** At present, there are three evaluation metrics that are used to measure the quality of video synopsis models. They are the metrics of collision cost (*CC*), temporal consistency (*TC*), and condensation ratio (*CR*). In this paper, we define another evaluation parameter in order to measure the spatial utilization of frames in the synopsis video and we called it the frame compact ratio (*FCR*).

*Collision Cost (CC):* In a condensed video, when a collision occurs between two tubes, the collision cost is calculated by counting the number of pixels within the collision areas. Usually, the collision cost is an important component of the energy function in a video synopsis model, as illustrated in Equation (Equation 2).

*Temporal Consistency (TC):* The temporal consistency metric is used to measure whether the correlations between activities in the original video are preserved in the condensed video [28]. Therefore, the *TC* is defined as: (7)$$E_{t}\left(o^{*},p^{*}\right)=\left\{\begin{array}{cc} D(o,p)\cdot T(o,p), & t_{o}\cap t_{p}\neq \phi \\ 0, & t_{o}\cap t_{p}=\phi ,T^{\prime }\left(o,p\right)>0\\ \exp\left(∥t_{o^{*}}^{s}-t_{p^{*}}^{s}∥/\sigma \right), & \;\mathrm{otherwise}\; \end{array}\right.$$

*Condensation Ratio (CR):* The *CR* is expressed as the ratio of the total frames of the condensed video to the total frames of the original one, and it is used to measure the video compression efficiency by using the following formula:(8)$$CR=\frac{\#Frame{}_{synopsis}}{\#Frame_{original}}\times 100\%$$

*Frame Compact Ratio (FCR) FCR* is defined as follows
(9)$$FCR=\frac{\#Pixel_{foreground}}{\#Pixel_{total}}\times 100\%$$
where the ratio of the foreground pixels to the total pixels in the synopsis video is calculated to measure the spatial utilization of the video condensation. The frame compact ratio and collision cost provide important perspectives on the impact of the condensation process.

### 4.2. Parametric Analysis

Experiments were conducted to evaluate the influences of parameters, such as the tubes’ container capacity *P*, the upper bound *H*, and the lower bound *L* for the collision cost. As illustrated in Figure 4a, when the capacity of tubelet *P* increased, the frame compact ratio (FCR) also increased because of the growth of the number of tubes that could be accommodated in the tube buffer and the range of time tags for the selected tubes. However, when *P* was too large (e.g., P≥10), the trends of the FCR were not evident, since *P* was already saturated and the rearrangement of tubes does not need to consider the influence of *P* in this situation. On the other hand, the collision cost (CC) gradually decreased as the tubelet capacity *P* increased, since the tubes could be rearranged more sparsely.

Figure 4b shows that the FCR decreased and the CC increased when the lower bound *L* in Algorithm 2 increased. The reason for this was that a larger *L* meant that more tubes were classified into the case in which L<=RC[i]<H was satisfied. The solution of the optimization in this case did not change the time positions of the tubes. Hence, the rearrangement of the tubes was more concentrated in both the spatial and temporal domains. Instead, as shown in Figure 4c, the influence of the upper bound *H* on the frame compact ratio was not obvious, although the impact on the CC was similar to that of *L* because the change in *H* had little effect on whether a tube would be re-categorized as a case in which L<=RC[i]<H. This only affected the number of tubes in cases where RC[i]>H.

In order to evaluate the contribution of dynamic tube resizing in the online video synopsis, experiments were conducted by using the traditional online video synopsis method without a tube-resizing operation, and this was compared with our proposed condensation method via dynamic and adaptive online tube resizing. Figure 5 shows the results of the synopsis of Video2 from the dataset.

The images were both located in the 435th frame of the condensation videos from the two different methods. From Figure 5b, we can see that the sizes of the moving objects (e.g., the white car) were adaptively adjusted in order to avoid collisions. In contrast, the object size remained the same in the whole synopsis video from the traditional online video synopsis method, as illustrated in Figure 5b. Consequently, the frames in the condensation video that was generated by using the proposed method contained more useful information. Our method achieved a much higher frame compact ratio. The averaged results of the frame compact ratio (FCR), as well as the respective temporal consistency (TC) and collision cost (CC), are presented in Table 2. Note that the maximum number of activities were displayed in one frame with minimal collisions because of the adaptive calculation of the tube-resizing coefficients according to newly generated tubes. It was indicated that our method that generated synopsis videos via dynamic and adaptive tube-resizing operations achieved a significant improvement in the frame compact ratio (FCR), temporal consistency (TC), and collision cost (CC) in comparison with the traditional online video synopsis method without a tube-resizing operation, as shown in Table 2. Meanwhile, the correlations between the activities in the original video were well preserved in the condensed video.

### 4.3. State-of-the-Art Comparison

The results of comparisons on the datasets with state-of-the-art methods are presented in Table 3. Among these methods that were selected for comparison, the methods proposed by Li et al. in [28] and by Nie et al. in [31] are offline video synopsis methods. Another selected online video synopsis approach was that proposed by Ruan et al. in [24].

As shown in Table 3, the computational cost of the online method in [24] was the lowest because it generated a synopsis video without resizing operations. However, the experimental results showed that the proposed algorithm had good real-time performance too, since the proposed online method enabled tube rearrangement at the time of detection without any waiting before starting optimization. In our proposed online video synopsis framework, a suitable mapping result could be obtained to represent the proper size of an activity in each frame of the condensed video. Hence, in terms of the condensation ratio (CR), our method outperformed the previous state-of-the-art methods.

Offline video synopsis methods always try to find the global optimum for all generated tubes. Hence, they can have the best rearrangement of the activities in order to display them in a short time period and easily achieve good performance in terms of the frame compact ratio (FCR) and collision cost (CC). It is noted the method proposed by Nie et al. [31] had the best performance in terms of the FCR and CC. However, it is complicated, and more than one hour was required to generate a synopsis video, which is impractical for real-world scenarios. On the other hand, our online method achieved a performance that was comparable to that of another offline method by Li et al. [28], and the computational time was shortened to half, as indicated in Table 3. Compared with the online video synopsis method given in [24], our method was able to achieve better performance on all key metrics, with almost the same computational complexity.

To analyze the condensation results of different synopsis methods more intuitively, we displayed some frames from the synopsis videos that were generated using the original Video1. As shown in Figure 6, we had the following findings: (1) More related activities were grouped and displayed together in the synopsis video generated by our method and by the method of Nie et al. [31], providing a better understanding of the scene for the user; (2) more activities were rearranged in a frame of the synopsis video to achieve a high frame compact ratio for our method and the method of Li et al. [28]. Our method avoided many collisions because of the dynamic tube resizing. (3) The proposed method reduced the optical motion anti-facts due to the smooth term of the resizing coefficients that was introduced in the video condensation model.

## 5. Conclusions

In this study, a simple online video synopsis framework was formulated. The proposed framework does not require any user interaction or prior models; it can condense activities in an original video into a shorter period by simultaneously showing multiple actions, and it can do so in real time. Different optimization strategies are applied according to different collision situations in order to maintain a balance among the computational cost, condensation ratio, and collision cost. Resizing coefficients are determined adaptively and dynamically in the optimization procedure; then, a suitable mapping result can be obtained to represent the proper size of an activity in each frame of the short condensed video. Finally, the maximum number of activities can be displayed in one frame with minimal collisions. The proposed online video synopsis framework can achieve a condensed video with a low collision cost. However, collisions cannot be completely avoided, and this degrades the visual quality of some frames in the synopsis video. Meanwhile, temporal consistency will be considered in the tube rearrangement in a future study.

## Figures and Tables

**Figure 1 sensors-22-09046-f001:**
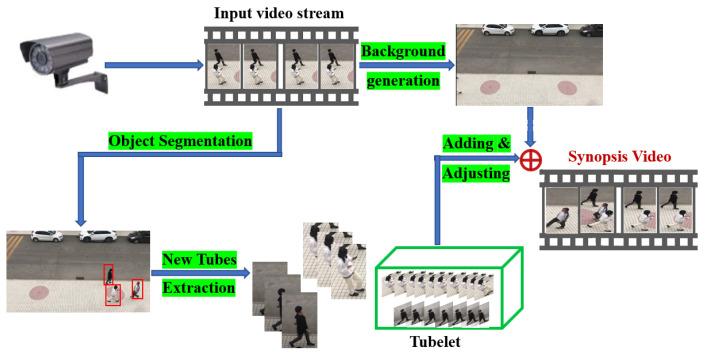
The online video synopsis framework.

**Figure 2 sensors-22-09046-f002:**
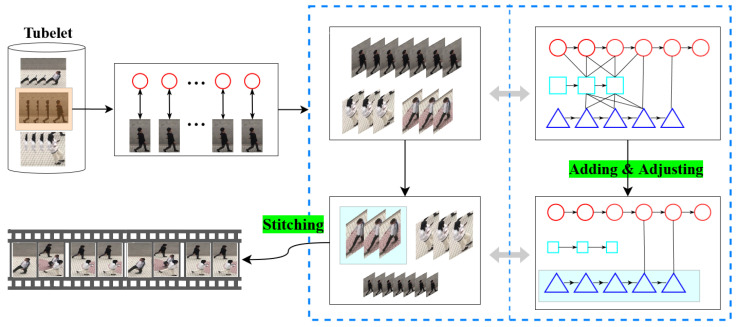
Overview of the adjustment phase in $G_{set}$ with a capacity of P=3.

**Figure 3 sensors-22-09046-f003:**
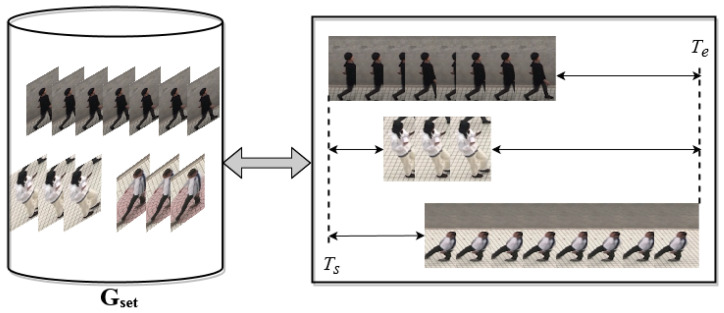
Tube addition phase. The length of $G_{set}$ is updated by $T_{e}-T_{s}$.

**Figure 4 sensors-22-09046-f004:**
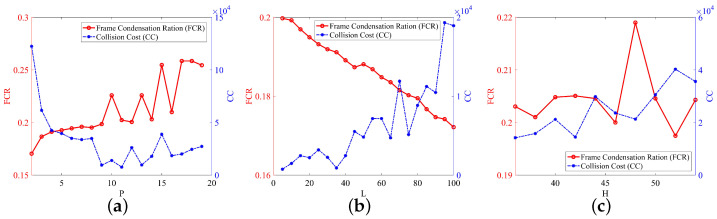
Frame compact ratio (FCR) and collision cost (CC) as a function of (**a**) container capacity *P*, (**b**) the lower bound *L*, and (**c**) the upper bound *H*.

**Figure 5 sensors-22-09046-f005:**
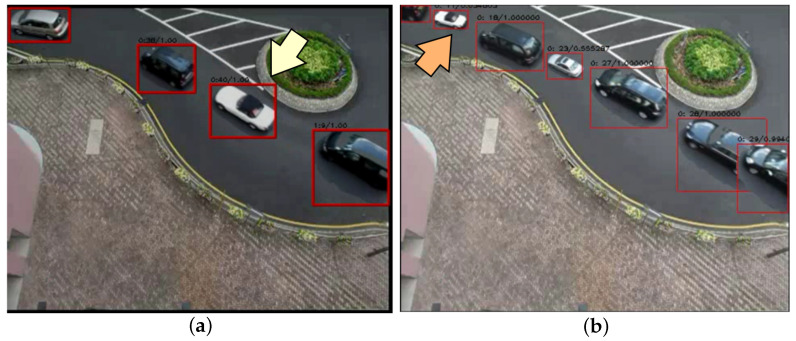
Frames in the synopsis video: (**a**) without a tube-resizing operation; (**b**) with dynamic and adaptive online tube resizing.

**Figure 6 sensors-22-09046-f006:**
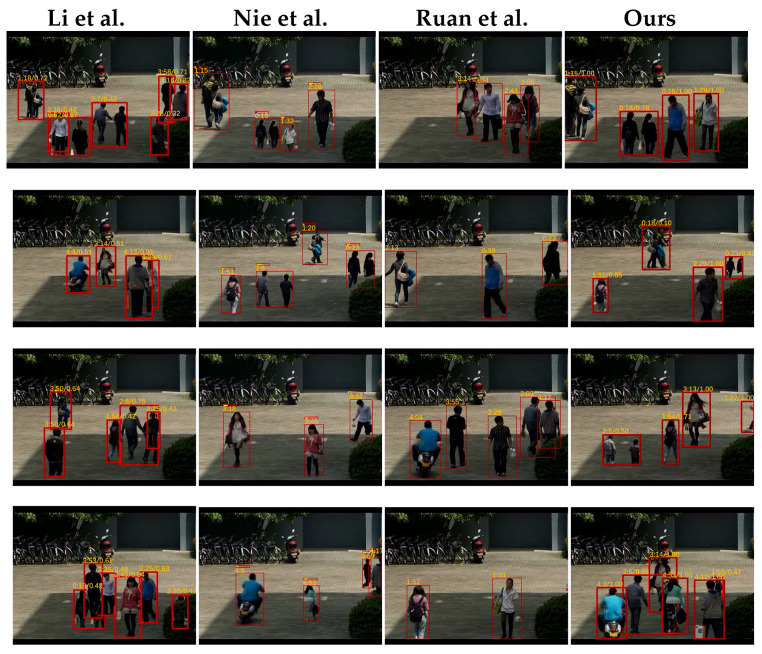
Frames from the synopsis videos that were generated by using different methods on Video1 [24,28,31].

**Table 1 sensors-22-09046-t001:** Information on the dataset.

Video Num	#Frame	#Object	Size
Video1	4514	14	368 × 276
Video2	3950	18	480 × 384
Video3	9390	42	368 × 276
Video4	970	11	720 × 480
Video5	4660	27	268 × 276

**Table 2 sensors-22-09046-t002:** Comparisons between the evaluation metrics of the results of the synopsis of Video2 from (a) the traditional method without tube resizing and (b) the proposed method with dynamic and adaptive online tube resizing.

Methods	FCR(%)	TC	CC
Without resizing	0.96	$1.25\times 10^{33}$	24,949
Ours	**1.30**	${8.73\times 10}^{23}$	**9113**

**Table 3 sensors-22-09046-t003:** Comparison of the averaged results with those of state-of-the-art methods on the whole dataset.

Methods	CR (%)	FCR (%)	TC	CC	Total Time (s)
**Li [28]**	10.0	2.10	$7.69\times 10^{9}$	$1.11\times 10^{6}$	69.23
**Nie [31]**	11.0	8.70	$2.67\times 10^{5}$	$2.82\times 10^{4}$	3953.33
**Ruan [24]**	10.0	1.10	$3.52\times 10^{16}$	$1.28\times 10^{8}$	36.73
**Ours**	9.00	1.80	$8.89\times 10^{7}$	$3.91\times 10^{6}$	39.76

## Data Availability

Not applicable.

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
