# Peer review of "Real-Time Video Synopsis via Dynamic and Adaptive Online Tube Resizing"

_sensors, 2022, doi:10.3390/s22239046_

Round 1

Reviewer 1 Report

The paper is written in good detail, though below are a few suggestions -

1. My primary concern is that the technical novelty should be clarified. Please specify in the introduction section, why it is important for the solution.

2. It should explain in more detailed comparative result why the proposed method is better than the previous methods mentioned in the paper.

3. There are a few grammatical errors, please review before final submission.

4. In the conclusion section, the limitations of the proposed method must be discussed by the authors and the same way to the future work 

5. As a suggestion, comparisons should be made with deep learning techniques.

Reviewer 2 Report

This very interesting paper describes an online video synopsis framework, which condenses activities into a shorter period, simultaneously showing multiple integrated tubes. Four evaluation metrics were used to measure the quality of video synopsis models and compare it to state-of-the-art methods.

The “related works” section provides sufficient references covering some of the main works in the area. Research design and analysis are properly developed.

The language requires some spell checking.

I think some questions should be addressed to the authors:

- The work by Barnich et al was used as a reference for background extraction and Fig 1 represents the structure, including a continuous background extraction. Authors should explain the real dynamics of background extraction.

- Regarding the processing speed, discussed in section “4.3. State-of-the-art comparison”, the authors claim that “online video synopsis methods are much faster than offline approaches” and this is often far from the truth, as it depends on the processing methods and assumptions. This statement should be clarified, please.
